# Degradation by Electron Beam Irradiation of Some Composites Based on Natural Rubber Reinforced with Mineral and Organic Fillers

**DOI:** 10.3390/ijms23136925

**Published:** 2022-06-22

**Authors:** Elena Manaila, Gabriela Craciun, Daniel Ighigeanu, Ion Bogdan Lungu, Marius Daniel Dumitru Grivei, Maria Daniela Stelescu

**Affiliations:** 1Electron Accelerators Laboratory, National Institute for Laser, Plasma and Radiation Physics, 409 Atomistilor Street, 077125 Magurele, Romania; elena.manaila@inflpr.ro (E.M.); daniel.ighigeanu@inflpr.ro (D.I.); marius.dumitru@inflpr.ro (M.D.D.G.); 2Multipurpose Irradiation Facility Center—IRASM, Horia Hulubei National Institute for R&D in Physics and Nuclear Engineering, 30 Reactorului Street, 077125 Magurele, Romania; ion.lungu@nipne.ro; 3Leather and Footwear Research Institute, National R&D Institute for Textile and Leather, 93 Ion Minulescu Street, 031215 Bucharest, Romania; dmstelescu@yahoo.com

**Keywords:** composites, natural rubber, precipitated silica, chalk, sawdust, hemp, electron beam, irradiation, degradation

## Abstract

Composites based on natural rubber reinforced with mineral (precipitated silica and chalk) and organic (sawdust and hemp) fillers in amount of 50 phr were obtained by peroxide cross-linking in the presence of trimethylolpropane trimethacrylate and irradiated by electron beam in the dose range of 150 and 450 kGy with the purpose of degradation. The composites mechanical characteristics, gel fraction, cross-linking degree, water uptake and weight loss in water and toluene were evaluated by specific analysis. The changes in structure and morphology were also studied by Fourier Transform Infrared Spectroscopy and Scanning Electron Microscopy. Based on the results obtained in the structural analysis, possible mechanisms specific to degradation are proposed. The increasing of irradiation dose to 450 kGy produced larger agglomerated structures, cracks and micro voids on the surface, as a result of the degradation process. This is consistent with that the increasing of irradiation dose to 450 kGy leads to a decrease in crosslinking and gel fraction but also drastic changes in mechanical properties specific to the composites’ degradation processes. The irradiation of composites reinforced with organic fillers lead to the formation of specific degradation compounds of both natural rubber and cellulose (aldehydes, ketones, carboxylic acids, compounds with small macromolecules). In the case of the composites reinforced with mineral fillers the degradation can occur by the cleavage of hydrogen bonds formed between precipitated silica or chalk particles and polymeric matrix also.

## 1. Introduction

Due to its specific physical and chemical properties, non-toxicity, renewability and not least due to the low price, the natural rubber is the most used elastomer worldwide. A common industrial practice used for the enhancement of physical and mechanical properties of a composite material based on natural rubber consists of the incorporation of fillers in the polymer matrix [1]. Many fillers are used now in the rubber industry but, from its first use and until now, the carbon black remains the most important reinforcing agent even if causes pollution and gives the rubber a black color [2]. The carbon black consists of small-size amorphous or paracrystalline carbon particles and is formed by the thermal decomposition of hydrocarbons, in the absence or presence of oxygen in substoichiometric quantities. Different commercial variants are available on the market as a function of the characteristics of the constituent particles as primary particle size, aggregate size and shape, porosity, surface area chemistry etc. [3]. Connected to the rubber industry needs, consistent research is carried out in order to identify other types of fillers, with different origin than petroleum and other color than black but with similar or better characteristics than the carbon black. Calcium carbonate, kaolin clay, precipitated silica, talc, barite, amorphous silica or diatomite were identified as being non-black fillers that increase the composite tear strength, adhesion to fibers or adjoining compounds, chemical compatibility or chemical resistance [4,5]. The technological progress has led to an increasing awareness of the needs related to the environmental protection [6,7,8,9]. Thus, composites reinforced with natural fibers, so-called biocomposites, were developed and play an increasingly important role in many fields as aerospace, automotive, packaging, furniture or construction and building materials [6,10,11,12,13]. The advantages of biocomposites use can be discussed in terms of physical/mechanical properties (low density, high strength and stiffness, stability) but also in terms processing advantages (fibers come from renewable sources, low energy expenditure than in the case of synthetic fibers production, low health risk during fiber processing, low emission of toxic vapors during the combustion of the depleted composite, flexibility during processing) [6,14,15,16,17]. Regardless of the filler type used for the reinforcement of the polymer matrix, the degradation of composites rubber based still represent a problem at the end of the life cycle, especially in the case of composites obtained by vulcanization in the presence of additives and vulcanization agents (sulfur, peroxides etc.) due to the strong cross-links that are formed between the rubber chains that make very slow the natural degradation process [18,19,20,21]. However, by applying an external energy (thermal, mechanical, photochemical, biological, chemical or from ionizing radiation) the existing three-dimensional networks in the vulcanized composites break resulting small fragments with lower molecular weight. The degradation process makes thus possible the physical recovery of so obtained wastes and their use as fillers in other elastomers-based composites [18,20]. By irradiating the polymers, both the branching and cross-linking of the molecular chains can be induced, leading to the increases of the molecular weight, but also the degradation that leads to the scission of the molecular chains. Thus, by degradation, the number and nature of the bonds are changed, oxidation reactions take place, and these types of reactions lead to a decrease in the molecular weight [18,22,23]. Chain branching, cross-linking and degradation or scission coexist during the irradiation process and are dependent on the material molecular structure and morphology, irradiation conditions (irradiation dose and dose rate) and irradiation atmosphere (rich or poor in oxygen). Polymers irradiation in the presence of oxygen (oxidative degradation) leads to the formation of free radicals that generate oxidized functional groups as carbonyl, peroxides, hydroperoxide, hydroxyl or carboxyl responsible with the chain scission and molecular weight decrease [24]. So, the presence of oxygen in the irradiation atmosphere sustains and increases the degradation process. Moreover, the higher the irradiation dose, the higher the amount and capacity of free radicals responsible for material degradation [24].

The purpose of the paper is to presents the results of some degradation studies made after irradiating the elastomeric composites based on natural rubber (NR) and four mineral and organic fillers. The elastomeric composites were cross-linked by means of peroxides in the presence of trimethylolpropane trimethacrylate (TMPT). Were used as fillers precipitated silica, chalk, sawdust and hemp in amount of 50 phr. The irradiation was performed at the National Institute for Laser Plasma and Radiation Physics—Electrons Accelerators Laboratory using the ALID 7 electron accelerator of 5.5 MeV. The working dose rate was of 3.3 kGy/min in order to accumulate doses between 150 and 450 kGy. Materials were tested before and after irradiation in order to highlight the degradation. Thus, the composites mechanical characteristics, gel fraction, cross-linking degree, water uptake and weight loss in solvents were evaluated by specific analysis. The changes in structure and morphology were also studied by Fourier Transform Infrared Spectroscopy and Scanning Electron Microscopy. Possible mechanisms specific to degradation are proposed.

## 2. Materials and Methods

### 2.1. Materials and Samples Preparation

In Table 1 are presented the properties of raw materials used for the elastomeric composites obtaining.

In Table 2 are presented the recipes that were the basis of the obtaining of elastomeric materials further used in degradation by irradiation studies. Mixtures based on natural rubber (NR), maleic natural rubber (NR-gAM) and fillers were made by means of blending technique on a laboratory roll as in previous studies [1]. The maleated natural rubber (NR-g-AM), was obtained by roll mixing the natural rubber (NR) with 5 phr of maleic anhydride for synthesis and 0.75 phr of Perkadox 40. The test specimen sheets of all composites were produced by compression molding at 165 °C for 24 min, according to the SR ISO 3417/1997. The optimum curing time (t90) of all spercimens was evaluated using an oscillating disk Monsanto rheometer [1] and the results obtained for control, NR-U, NR-C, NR-S and NR-H were 12.47, 5.45, 9.34, 6.22 and 6.81, respectively.

Figure 1 shows the grafting mechanisms of maleic anhydride on the natural rubber chain and the obtaining of maleated natural rubber [30,31]. Possible cross-linking/grafting ways of cellulose from wood sawdust or hemp on the natural rubber chains are presented in the Figure 2a,b. The natural rubber and cellulose join together through the C-C and C-O-C bonds [32].

Possible interactions between the maleated natural rubber and mineral fillers (precipitated silica and chalk) are presented in Figure 3a–c [33,34,35].

### 2.2. Samples Irradiation

The elastomeric composites were subjected to electron beam irradiation at the irradiation doses of 150, 300 and 450 kGy with the purpose of degradation. Irradiation was performed in atmospheric conditions and at room temperature of 25 °C using the electron linear accelerator of 5.5 MeV, ALID 7 from the National Institute for Laser, Plasma and Radiation Physics, Magurele, Romania. The nominal values of the electron beam (EB) parameters were as follows: energy of 5.5 MeV, peak current of 130 mA, output power of 134 W, pulse duration of 3.75 μs and pulse repetition frequency of 50 Hz [18,36]. The irradiation process performance depends on the rigorous control of the irradiation dose and dose rate. In our experiments, the process dose rate was of 3.3 kGy/min. The primary standard graphite calorimeter was used for radiation dosimetry. In order to ensure equality between the doses at the entrance and exit of the irradiated sample and for an efficient use of the electron beam, the penetration depth was calculated according with the following equation [18,36].
(1)E=2.6⋅t⋅ρ+0.3
where *E* (MeV) is the electron beam energy, *t* (cm) is the sample thickness, and *ρ* (g·cm^−3^) is the sample density (in our case, 1 g·cm^−3^). The proper thickness of samples subjected to EB irradiation was calculated as being of 20 mm [18,36].

In order to prevent the samples heating over 80 °C, the irradiation process was made in sessions of 50 kGy alternating with rest for accelerator cooling. The irradiation–rest–irradiation cycle was repeated until the mentioned irradiation doses were accumulated.

### 2.3. Laboratory Tests

#### 2.3.1. Mechanical Characteristics

Tensile strength (σ) and specific elongation (ε) have been determined using the Material Testing Machine ProLine Z005 from Zwick-Roell, Ulm, Germany, equipped with a 5 kN cell force, in accordance with DIN EN ISO 527-1 (ISO 37: Rubber, vulcanized or thermoplastic—Determination of tensile stress-strain properties, Sixth Edition, 11-2017). The dumb-bell test pieces were of type 4 having the overall length of 35 mm, length of narrow portion of 12 ± 0.5 mm, test length of 10 ± 0.5 mm, thickness of narrow portion of 2 ± 0.1 mm and width of 2 mm. The test speed was of 200 mm/min. Hardness was measured using a hardness tester according to ISO 7619-1/2011 on samples having the thickness of 6 mm. All measurements were carried out for minimum five times.

#### 2.3.2. Gel Fraction and Cross-Link Density

In order to determine the gel fraction of the cross-linked products (gel content), the solvent extraction method using toluene was used [18,36]. The tests were performed at room temperature of 23 ± 2 °C on circular samples of 15 mm diameter and 2 mm thickness according to the SR EN ISO 20344/2004 standard. Samples weighed and swollen in toluene for 72 h, were dried in air for 6 days and in a laboratory oven at 80 °C for 12 h for completely removing of the solvent. Samples were finally, reweighed and the gel fraction was calculated as follows:(2)$$\mathrm{Gel}\;\mathrm{fraction}=\frac{m_{s}}{m_{i}}\times 100$$
where *m_s_* and *m_i_* are the weights of the dried samples after and before swollen [37,38].

The cross-link density (*ν*) was determined based on equilibrium solvent-swelling measurements in toluene and by applying the modified Flory-Rehner equation for tetra functional networks. Samples with thickness of 2 mm were initially weighed (*m_i_*) and immersed in toluene for 72 h at room temperature. After 72 h, the swollen samples were extracted from toluene, wiped with paper towels for solvent excess removal and weighed (*m_g_*) in special ampoules to avoid toluene evaporation during weighing. All samples were dried in air for 6 days and in a laboratory oven at 80 °C for 12 h for completely removal of the solvent. Finally, the samples were weighed for the last time (*m_s_*) and the volume fractions of polymer in the samples at equilibrium swelling (*ν*_2*m*_) were determined from swelling ratio *G*, as follows:(3)$$\nu_{2m}=\frac{1}{1+G}$$
where,
(4)$$G=\frac{m_{g}-m_{s}}{m_{s}}\times \frac{\rho_{e}}{\rho_{s}}$$
and *ρ_e_* and *ρ_s_* are the densities of elastomeric samples and solvent (0.866 g/cm^3^ for toluene), respectively.

The hydrostatic weighing method, according to the SR ISO 2781/2010 was used for the elastomers densities determining. The volume of samples is the difference between the sample weight in air and in a liquid of a known density divided by the density of this liquid. The cross-link density, *ν*, was determined from measurements in toluene, using the Flory–Rehner relationship:(5)$$\nu =\frac{Ln(1-\nu_{2m})+\nu_{2m}+\chi_{12}\nu_{2m}^{2}}{V_{1}\left(\nu_{2m}^{1/3}-\frac{\nu_{2m}}{2}\right)}$$
where *V*_1_ is the molar volume of solvent (106.5 cm^3^/mol for toluene), *ν*_2*m*_ is the volume fraction of polymer in the sample at equilibrium swelling, and *χ*_12_ is the Flory–Huggins polymer–solvent interaction term [37,38]. The value of Flory-Huggins polymer-solvent interaction term ($\chi_{12}$) for the natural rubber—toluene system was of 0.393 [2,38].

#### 2.3.3. Cross-Link and Chain Scission Yields

The quantitative evaluation of cross-linking and chain scission yields of irradiated samples was carried out by plotting the *S* + *S*^1/2^ vs. 1/absorbed dose (*D*) from the Charlesby–Pinner equation [39,40]:(6)$$S+\sqrt{S}=\frac{p_{0}}{q_{0}}+\frac{1}{\alpha P_{n}D}$$
where, *S* is the sol fraction (*S* = 1 − gel fraction), *p*_0_ is the degradation density—the average number of main chain scissions per monomer unit and per unit dose, *q*_0_ is the cross-linking density—proportion of monomer units cross-linked per unit dose, *P_n_* is the number averaged degree of polymerization and *D* is the radiation dose in kGy.

The increasing of *p*_0_/*q*_0_ ratio is associated with the prevalence of scission reaction against cross-linking, being well known the fact that *p*_0_ represent the degradation degree (average number of chain scissions per monomer unit) and *q*_0_ the cross-link density [18].

#### 2.3.4. Rubber-Filler Interaction

The extent of interaction between natural rubber and filler was analysed using the Kraus equation [41]:(7)$$V_{ro}/V_{rf}=1-m\frac{f}{1-f}$$
where, *V_ro_* and *V_rf_* are the volume fractions of the rubber in the vulcanized rubber and in the filled swollen sample, respectively, *f* is the volume fraction of filler and *m* the filler—polymer interaction parameter. The volume fraction of rubber in the swollen sample *V_rf_*, was calculated using the following equation [41]:(8)$$V_{rf}=\frac{\left[\left(D-FT\right)/\rho_{r}\right]}{\left[\left(D-FT\right)/\rho_{r}\right]+\left(A_{0}/\rho_{S}\right)}$$
where, $\rho_{r}$ is the density of natural rubber with and without fillers, $\rho_{s}$ is the density of solvent, *D* is the de-swollen weight of the test specimen (dry weight), *F* is the weight fraction of the insoluble components, *T* is the weight of the specimen and *A*_0_ the weight of the absorbed solvent at equilibrium swelling.

#### 2.3.5. Water and Toluene Uptake and Weight Loss

The solvents uptake in irradiated and non-irradiated elastomeric composites was evaluated as in previous works [18,36], in accordance with ISO 20344/2011, by immersion in water and toluene at room temperature of 23 ± 2 °C, until the samples no longer absorbed solvents (uptake at equilibrium). The water and toluene uptake in composites as well as the weight losses by immersion in them were evaluated based on the results obtained for gel fraction and cross-linking degree. The uptake at equilibrium in toluene was reached after 6 days and in water after 100 days. After the equilibrium was reached, samples have been dried in air for 6 days then in laboratory oven at 80 °C for 72 h [18].

#### 2.3.6. Structural Investigations by Fourier Transform Infrared Spectroscopy

The elastomeric composites structure before and after irradiation, was investigated by Fourier Transform Infrared Spectroscopy (FTIR) technique using the Spectrum 100 instrument (Perkin Elmer, Waltham, MA, USA). The FTIR spectra were acquired in ATR mode in the range of 4000–650 cm^−1^ with a 50 scans/sample and a resolution of 4 cm^−1^.

#### 2.3.7. Morphological Investigations by Scanning Electron Microscopy

The elastomeric composites’ surfaces before and after irradiation were examined by Scanning Electron Microscopy (SEM) technique using the FEI/Phillips scanning electron microscope (Hillsboro, OR, USA). Samples fractured in liquid nitrogen and sputtered with gold palladium, were placed in aluminum mounts and scanned at an accelerating voltage of 30 kV.

## 3. Results and Discussion

### 3.1. Mechanical Characteristics

The influence of the filler type and irradiation dose on hardness, tensile strength and elongation of NR-H, NR-S, NR-U, NR-C composites and control sample were evaluated and the results are presented in Table 3 and Figure 1a–c. Associated with the degradation process, the hardness (as a measure of the material resistance to an applied deformation) must increase while the tensile strength (the maximum elongation supported by the material until it breaks) and the elongation at break (the resistance of the material to shape changes until the formation of cracks) must decrease.

As it can be seen from Table 3 and Figure 1a–c the mechanical properties that have been investigated were differently affected by the irradiation. The presence of fillers in the composites, also contributed to the obtained results. The values in Table 3 showed that mechanical properties are modified (improvement) for non-irradiates composites) simply by the filler addition. By irradiation of control samples (NR), the hardness has increased from 45.33 to 48.67 °ShA and tensile strength and elongation at break has decreased from 6.23 to 1.77 MPa and from 89.75 to 17.4%, respectively. Both tensile and elastic properties specific to degradation were obtained for all types of composites but better for samples reinforced with mineral fillers, precipitated silica (NR-U) and chalk (NR-C). An explanation can be associated with the filler particle sizes: the larger they are (as in the case of hemp and sawdust) the more agglomeration tendency that leads to a defective interface with the polymer matrix. In addition, the appearance of gaps in the material due to the difference in size particles can be a possible explanation [18,42,43,44,45].

The effect of EB irradiation treatment was evaluated in terms of mechanical properties variation yield of the irradiated versus non-irradiated samples and the tendency is presented in Figure 1a–c. At the irradiation dose of 450 kGy, the following hardness variation yields were registered (Figure 1a): increases of 7.36%, 20.43% and 15.42% for control sample, NR-S and NR-C, respectively, and decreases of 0% and 5.74% for NR-H and NR-U, respectively. From all four types of composites that have been tested, the most sensitive in terms of hardness to radiation action were those reinforced with sawdust and chalk. Tensile strength variation yields have presented declining trends by the irradiation treatment, excepting NR-S at all irradiation doses, NR-U and NR-C at 150 kGy. It must be pointed out the reduction of 80.06% obtained for NR-C at 450 kGy (Figure 1b). As it can be seen from Figure 1c, the reduction of elongation at break variation yields can be associated with the degradation process, especially in the case of irradiation at 450 kGy of control sample (80.61%), NR-H (81.31%) and NR-C (77.58%). Generally, the reinforced rubber materials exhibit increased mechanical strengths than unfilled rubbers, hardness being one of the most widely measured property connected with the material elasticity and usually used as a benchmark for assessing the degree of vulcanization [42]. In addition, in the case of tensile strength were registered different behaviors of tested materials as a function of filler type and irradiation dose. At the lowest irradiation dose of 150 kGy, NR-C and NR-U showed important increasing of tensile strength versus non irradiated samples. The effect can be connected with a better interaction between the polymeric matrix and filler. The polymeric chains movement is restricted and the adhesion between the polymeric matrix and filler is improved by irradiation [32,46,47]. A continuation of potentially unfinished cross-linking reaction by the peroxide curing can take place by irradiation [18]. The increasing of the irradiation dose leads to the cleavage bonds appearance, responsible for the reduction of mechanical properties. The irradiation experiments took place in the presence of oxygen that generates free radicals, oxidizing functional groups (carbonyl, peroxides, hydroxyl) and, as a consequence, cleavage reactions that coexists with those of cross-linking. The higher the irradiation dose, the higher the potential for cleavage and oxidation reactions [24]. The filler addition and irradiation dose increasing have led to the decreasing in tensile strength and elongation at break (excepting NR-S), phenomena associated with degradation. All the composites has become harder, less resistant at break and ductile due to the restriction of the polymeric chains free movement, meaning that the degradation due to irradiation has been highlighted [18,48,49,50]. However, the least degradable composite in electron beam is the one reinforced with precipitated silica (NR-U).

### 3.2. Sol-Gel and Cross-Link Density

The gel fraction and cross-link density are used to measure the mass fraction of the network resulted in the cross-linking process and the number of cross-links per unit volume in the same network, respectively. The influence of filler type and irradiation dose on gel fraction and cross-link density of composites discussed is presented in Table 4 and Figure 2a,b.

As seen in Table 4, the gel fraction values of non-irradiated reinforced samples are comparable with those of control samples. The fillers addition changed the cross-link density from single to more than two and a half times of non-irradiated samples. By applying the irradiation treatment, the cross-link density of all samples has increased, especially of samples reinforced with natural fillers. It looks similar to the natural fibers generate stronger reinforcing effects than organics.

The fillers addition led to semnificative modifications of variation yields. The largest increase of gel fraction variation yield (Figure 2a) was observed at the irradiation dose of 150 kGy: 0.93% for NR-H, 0.88% for NR-S, 1.46% for NR-U and 0.87% for NR-C. Further increasing of the irradiation dose led to a decrease in the gel fraction variation yield for all mixtures, but this remained higher than in the non-irradiated sample case. Irradiation dose and filler type have different effects on the cross-linking degree variation yield (Figure 2b). So, in the case of NR-H, it registered an increase of 4.23% only at the irradiation dose of 150 kGy. Further increasing of irradiation dose generates values under 8.1238 × 10^−4^ mol/cm^3^ registered for the control sample. From all mixtures, the cross-linking degree of NR-S increases with the irradiation dose: 38.13%, 27.16% and 23.93% at 150 kGy, 300 kGy and 450 kGy, respectively.

It is easy to observe the different behavior in radiation field of composites reinforced with mineral fillers (precipitated silica and calcium carbonate) to organic fillers (hemp and sawdust). Both hemp and sawdust contain cellulose, hemicellulose, lignin, wax, water etc. The unidirectional cellulose microfibers formed are reinforcing elements in polymer matrix/cellulosic fiber composite. This effect is sustained by the high values of cross-linking degree obtained for irradiated and non-irradiated NR-H and NR-S, probably due to an increase in stiffness of the composite due to the presence of organic fillers and degradation also [51]. If the macromolecular chains have lower mobility, the elasticity decreases (elongation, elongation at break) and stiffness (hardness) increases [51].

The natural calcium carbonate is in the form of particles with low aspect ratio, low surface area and poor adhesion with polymeric matrices [4]. Precipitated silica consists of a three-dimensional network of coagulated primary silica particles [4]. The poor interaction/adhesion between the mineral fillers and natural rubber compared with organic fillers is observed especially from the results obtained in cross-linking degree evaluation, but from the mechanical properties also. The cross-linking degree of NR-U and NR-C is less affected by the irradiation dose increasing than of NR-H and NR-S.

The extent of interaction between natural rubber and filler was analyzed using the Kraus equation and the results are listed in Table 5. In order to appreciate the interaction between the filler and the polymer matrix as being good, the values obtained for *V_ro_*/*V_rf_* must be under the unit (*V_ro_*/*V_rf_* < 1).

The ratio *V_ro_*/*V_rf_* provides information about the restriction of swelling of the rubber matrix due to the presence of fillers [41]. The more and more reduced values of *V_ro_*/*V_rf_* ratio are associated with the enhanced adhesion between filler and rubber, according to the Kraus theory and Kraus equation. The higher the adhesion between the polymeric matrix and the filler material, the lower the formation of voids in the composite material, and thus the lower the solvent absorption [41]. From Table 5 it can be observed that the equilibrium solvent uptake in the composites is dependent on the filler type. The toluene uptake at equilibrium for non-irradiated samples increases in the following order: NR-H < NR-S < NR-U < NR-C. The same order of decreasing was also found for *V_rf_*, so the ratio *V_ro_*/*V_rf_* increases alos in the same way, since *V_ro_* is constant. The composites under discussion are vulcanized with dibenzoyl peroxide at 165 °C. NR-H and NR-S present the best adhesion between rubber and matrix due to the reaction of vulcanizing agent with the organic fibers. That leads to the reduction of the hydroxyl group content responsible with the strong hydrophilic character of the natural fibers, improving thus the compatibility and the properties of composites [51,52].

### 3.3. The Cross-Link and Chain Scission Yields

In order to determine the proportion between the cross-linking and scission, plots of S + S^1/2^ vs.1/absorbed dose (1/D) from the Charlesby-Pinner equation were carried out [39,40]. The results are presented in Figure 3 from which was calculated the ratio *p*_0_/*q*_0_, and the obtained values are presented in Table 6.

The results presented in Table 6 show that the ratio *p*_0_/*q*_0_ is dependent, as it was expected, on filler type in the composite. This ratio increasing is associated with the increasing of scission reactions number, associated with the degradation, over the cross-linking reactions number; *p*_0_ is the degradation degree and *q*_0_ is the cross-linking degree. The order in which this ratio grows is: control < NR-C < NR-S < NR-H < NR-U, which shows that the natural rubber is less degraded by electron beam irradiation than any other composites under discussion. 

Natural rubber cross-linking by means of dibenzoyl peroxide is initiated by the decomposition of peroxide in primary radicals that start the cross-linking process. These radicals participate in propagation reactions forming stabile (acetone and diacetyl benzene) and instable species (of CH_3_•form), the latter continuing the natural rubber cross-linking reaction. The natural rubber macroradicals binds to the C=C bond of polyfunctional monomer TMPT. In addition, the radicals formed by decomposition of peroxide react with the hydroxyl groups or hydrogen from cellulose or lignin that are the main components of hemp and sawdust, thus occurring the grafting of natural rubber on these. By cross-linking of composites based on natural rubber and organic filler (hemp and sawdust) by means of peroxyde, C-C and C-O-C bonds are formed [53]. The increasing of *p*_0_/*q*_0_ ratio of NR-S and NR-H as against the control sample shows that, by electron beam irradiation, the degradation reactions number increases due to the C-C and C-O-C bonds splitting [54]. On the other hand, the reinforcement of elastomers with mineral particles is strongly dependent on their dimensions. The strongest interaction between the mineral filler (silica, chalk) and natural rubber is easily produced if the particle size is below 100 nm. A so-called semi-reinforcement takes place if the particles dimensions are between 100 and 1000 nm. Over this, the reinforcement capacity decreases more, fillers becoming detrimental for the composite structure [55,56]. Besides the particles’ dimensions, their chemical structure is of a great importance. Silica has a number of hydroxyl groups on its surface that determine the formation of strong hydrogen bonds in mixtures with polar polymeric materials. The filler particles become aggregates, agglomerates and clusters that prevent the silica dispersion in the non-polar polymeric matrix of natural rubber [55,57]. The adhesion between rubber and silica is poor in this case. 

Chalk exhibits the same deficiency of polar surface chemistry. It is difficult to moisten and disperse in the natural rubber, so the interaction with the rubber matrix is weak [58,59]. Thus, for a better interaction between precipitated silica or chalk and natural rubber matrix, the maleated natural rubber (NR-g-AM) was used. The results presented in Table 5 confirm the poor adhesion between the mineral fillers and natural rubber matrix even in the case of non-irradiated samples. Silica and chalk have large interface area due to their large particle sizes, this aspect influencing the degradation processes by irradiation [60]. These are reasons for high *p*_0_/*q*_0_ ratios, especially in the case of NR-U.

### 3.4. Water and Toluene Uptake Tests and Weight Loss in Solvents

The water and toluene uptake at equilibrium in irradiated and non-irradiated composites was evaluated by immersion in both solvents at room temperature (23 ± 2 °C). Uptake at equilibrium in toluene was reached after 6 days and in water after 50 days. After the reaching equilibrium, samples have been dried in air for 6 days then in laboratory oven at 80 °C for 72 h. These tests were carried out in order to demonstrate the influence of high hydrophilic polar groups from fillers on degradation by irradiation.

The uptake of solvents in polymeric materials is achieved by absorption and diffusion phenomena and is strongly dependent on the polymeric matrix and filler’s ability to provide pathways for the solvent to progress in the voids randomly formed. As much the number of voids is lower as much the solvent amount absorbed is lower, too. The equilibrium swelling is a good technique that can be used to evaluate the solvents penetration in elastomeric composites and the adhesion between the polymeric matrix and filler, being known that in a polymeric composite, solvents can be absorbed by both polymeric matrix and filler. The higher the adhesion between the matrix and filler, the lower is the amount of solvent that can penetrate the composite material [60]. The results of water and toluene uptakes at equilibrium, as well as the samples mass loss variation in these solvents, are presented in Table 7 and Figure 4a,b.

It was expected that the composite materials reinforced with hemp and sawdust to exhibit the highest percentages of water uptake due to the strong hydrophilic character of natural fibers. As seen in Table 7, the water uptake of non-irradiated NR-H (12.13%) is higher than of NR-S (6.64%) due to the higher content of cellulose. The number of OH groups is high as the cellulose content is also high. The OH groups attract water molecules and the water uptake increases. By electron beam irradiation at a dose of 450 kGy, the percentages of water uptake at equilibrium have increased for control sample, NR-S and NR-C, and decreased for NR-H and NR-U. The results obtained in case of composites reinforced with mineral fillers may be explained by the presence of the maleinated natural rubber (NR-g-AM) that acts as a plasticizer and favors a good interaction at the interface between the hydrophobic matrix and hydrophilic filler. Although water absorption is low for NR-C and NR-U, a decrease with the irradiation dose for NR-U and an increase for NR-C was observed. The result obtained for NR-C can be explain by the agglomeration of the filler that could led to the formation of so called “interfacial voids” at the particle-polymer interface with the consequence of water uptake increasing [6]. The decreasing of water uptake for NR-U with the irradiation dose increasing indicates that the surface of precipitated silica becomes less hydrophilic, i.e., the silanol on the silica surface has degraded. The free radicals formed by irradiation can recombine or diffuse to the surface, making it reactive and accelerating degradation reactions in the vicinity of the filler surface. Thus, precipitated silica becomes more hydrophobic and interacts more easily with the polymeric matrix in its vicinity forming covalent bonds that increase the adhesion between them [60]. As seen in Table 7, the biggest weight loss in water was registered in the case of composites reinforced with hemp, NR-H, and irradiated at 300 and 450 kGy. Unexpected results were obtained for irradiated NR-S (the decreasing of weight loss in water with the irradiation dose increasing). Correlating these results with those obtained for cross-linking degree which decreased with the irradiation dose for NR-H but increased for NR-S, it can be said that degradation effect in NR-H is more pronounced than in NR-S. The effect of EB irradiation treatment was evaluated in terms of weight loss variation yield of the irradiated versus non-irradiated samples in water and toluene and the tendency is presented in Figure 4a,b. The weight loss variation yields in water of control sample, NR-U, NR-C and NR-S at 150 and 300 kGy were negative while of NR-H and NR-S at 450 kGy were positive. These results, together with those previously obtained, show that the electron beam irradiation led to the increasing of cross-linking degree (Figure 2b) and rubber-filler interaction (Table 5), especially for the irradiation dose of 450 kGy. It can be said that the addition of minerals, precipitated silica and chalk, in the natural rubber matrix led to the obtaining of more resistant and stabile composite materials in radiation field than the addition of organics, hemp and sawdust. Quite opposite to the results obtained in the case of water uptake at equilibrium are the ones obtained in toluene. As seen in Table 7, all non-irradiated reinforced samples exhibited lower uptakes in toluene compared to control sample. Since both toluene and natural rubber are non-polar, the solvent is absorbed especially by the rubber matrix. In this case, the absorption due to the filler hydrophilicity can be neglected. In addition, some uncross-linked chains from the polymeric matrix are dissolved by toluene. The decrease of toluene absorption in the composites shows an increase of the cross-linking degree, but also an increase of the interaction between the elastomer and filler [61,62,63]. As much as the cross-linking degree is high as much as the toluene absorption is low. Only in NR-H the toluene uptake increases up to 130.24% due to the irradiation at 450 kGy. This result suggests that NR-H degrade easier by irradiation. In contrast with NR-U and NR-C samples in which the toluene absorption in the irradiated samples is comparable to that in the non-irradiated samples, NR-S showed a decrease in toluene absorption as the irradiation dose has increased, demonstrating an increasing of cross-linking degree by irradiation. By increasing the irradiation dose, scissions of grafting and cross-linking bonds have occurred and some compounds with small molecules that have been formed are removed in toluene. Correlating the results with those presented in Figure 2b for the same NR-U and NR-C, it can be said that the dispersion of mineral fillers in the composites is superior to the one of organic fillers [64].

### 3.5. Structural Investigations by Fourier Transform Infrared Spectroscopy (FTIR) Technique

During electron beam irradiation, three types of reactions occur simultaneously: cross-linking, branching of the polymer chains and breaking of polymeric chains and grafting bonds. Methyl or tert-butyl groups are formed by homolithic or heterolytic dissociation. During irradiation there are also oxidation reactions that lead to the formation of peroxides, alcohol groups, carboxylic groups, peroxide groups [18,65]. Bands associated degradation by irradiation such as alcohols, aldehydes, ketones, carboxylic acids, esters and hydroperoxides must be identified in the composites under discussion [18,66,67]. The irradiation dose increasing should result in changes in the absorption intensity of the bands associated with the degradation products thus identified. The FTIR results obtained for the samples before and after irradiation are presented in Figure 5a–d.

All the composites under discussion present broad bands between 3800 and 3000 cm^−1^ due to the presence of proteins from natural rubber [18]. Radiation treatment led to an absorption increasing in this area, especially between 3359 and 3316 cm^−1^, followed by a decreasing due to the hydroxyl groups (-OH) generated by oxidative degradation [18,66,68,69]. The band at 3359–3316 cm^−1^ is more pronounced in the case of NR-H and NR-S and is characteristic for stretching vibration of the hydrogen bonded hydroxyl group in polysaccharides [70]. All the composites under discussion present bands at 3037–3034 cm^−1^ that correspond to the stretching vibration of =CH- in the CH=CH_2_ group from natural rubber that is also associated with the degradation process [18,71]. The saturated aliphatic sp^3^ C–H bonds (ν_as_ (CH_3_), ν_as_ (CH_2_), and ν_s_ (CH_2_)) attributed to the bonds in the alkyl groups were found between 2960 and 2850 cm^−1^ and, as the irradiation dose increased over 150 kGy, decreases in the intensities for these groups were registered [72,73,74]. Bands that have been found at 1739–1732 cm^−1^ in the non-irradiated samples correspond to fatty acids and their esters and may be assigned to carbonyl group (-C=O) from ketone (R_2_C=O) or aldehyde (RCOH) resulted from the oxidative degradation. Due to the irradiation treatment, decreasing in bands intensities for control samples, increasing for NR-S and NR-H and insignificant variations for NR-U and NR-C were found here [66,71,74,75]. Bands attributed to -C=C- stretching vibration in the NR structure, to hydroxyl stretching vibrations of absorbed water and to carboxylate or conjugated ketone (>C=O) that are oxidative products derived from degradation, were found between at 1658–1647 cm^−1^ and 1655–1651 cm^−1^ [18,71,76]. Bands that correspond to symmetric and asymmetric deformations of –CH_2_ and CH_3_ from natural rubber have been found at 1448–1445 cm^−1^ and 1376–1375 cm^−1^, respectively [18,77]. The modification of these bands due to irradiation shows that the degradation of composite materials also occurs through the oxidation of methyl terminal groups with the formation of degradation products such as the formation of carboxylic acid derivatives group, the hydration of double bond to tertiary alcohol (aldehyde and ketones) and the cleavage of the internal bond resulting in polymeric molecular chain breakage [74,78,79]. Another sign of degradation due to irradiation is the decreasing of CH_2_ and CH_3_ vibration at 1360–1230 cm^−1^ [80]. Impurities from the natural rubber (proteins, lipids, amino acids, peptides) are instable and easily degradable by irradiation. The bands corresponding to their degradation were found at 1248–1236 cm^−1^ and 1156–1154 cm^−1^ [18,77,81,82]. At 1150–1030 cm^−1^, the non-irradiated samples show absorptions that are due to the multifunctional monomer used in the cross-linking process. Absorptions in this range show the presence of the methyl methacrylate (MMA) group in TMPT, due to the –C–O– fragment in the ester functional groups of methyl methacrylate [83,84]. The increase of the irradiation dose leads to the decrease of the intensity of these bands due to the cleavage of the crosslinking reactions in the presence of the polyfunctional monomer, i.e., the degradation of the composite materials occurs.

Absorption that has been found between 1248 and 1236 cm^−1^ in all samples’ spectra is characteristic to the number of cis-1,4 double bonds in the polyisoprene chain due to CH_3_ rocking (cis- polyisoprene), CH_3_ and C-CH_3_ stretching (trans- polyisoprene and cis- polyisoprene) and =CH bending and wagging (trans- polyisoprene) [80]. Increased absorption at 874–872 cm^−1^ in all irradiated samples spectra is due to O-O stretching in tertiary hydroperoxides which are amongst the first degradation products that appear. In addition, by irradiation dose increasing, modified absorptions at 1030–1020 cm^−1^ and 750–710 cm^−1^ were found and are assigned to the C-O stretching mode and of O-H deformation from unsaturated primary alcohols, both being degradation products [80].

In addition to the natural rubber matrix, the filler materials also degrade. Thus, the composites reinforced with organic fillers, NR-S and NR-H, have generated modified absorptions at 3400–2800 cm^−1^ attributed to the C-H and OH groups from degraded cellulose and hemicellulose. In the regions 1658–1647 cm^−1^ and 1551–1537 cm^−1^ were found increase in absorptions assigned to stretching vibrations of aromatic C=C and stretching vibrations of conjugated C=O, respectively. Symmetric and asymmetric bending of CH_3_ groups from degraded cellulose and hemicelluloses have generated decreased absorptions due to the irradiation dose increasing in the region 1376–1375 cm^−1^. Bands between 1248 and 1236 cm^−1^ assigned to stretching vibrations of C-O in xylan, syringyl ring and lignin showed decreased intensities in irradiated samples. Absorption bands at 1160–1030 cm^−1^ attributed to the stretching and asymmetric vibrations of C-O, C-C and C-O-C from cellulose and hemicelluloses have showed decreases with the irradiation dose increasing. All changes highlighted in the spectra of the irradiated NR-S and NR-H composites, prove the degradation of cellulose, hemicellulose and lignin from the organic fillers’ hemp and sawdust [85]. The same discussion will be carried out forward for the composites reinforced with mineral fillers, NR-U and NR-C. The analysis of NR-U samples highlighted the absorption at 3343–3331 cm^−1^ due to the OH stretching from the silanol group. Absorptions at 1658–1647 cm^−1^, 1100–1080 cm^−1^ and 827–816 cm^−1^ are assigned to the stretching vibration of O–H bond peak, asymmetric stretching of Si–O group and blending vibration of siloxane groups (Si–O–Si) [86]. Decreased absorption of these bands with the radiation dose increasing is a consequence of the degradation of the composite based on natural rubber reinforced with precipitated silica. The analysis of NR-C samples highlighted an absorption at 3378–3316 cm^−1^ due to the hydroxyl (–OH) functional groups of polymeric matrices. This peak confirms the surface modification of the filler [87]. Bands observed at 2516–2514 cm^−1^ and 1797–1793 cm^−1^ decrease with the irradiation dose increasing and are assigned to the doubly degenerate planar asymmetric stretching and doubly degenerate planar bending of CO_3_^2−^ [88,89]. Band at 1437–1436 cm^−1^ confirms structural changes in the molecular symmetry of the CO_3_^2−^ ion and corresponds to its asymmetric stretching mode. The changes that occur in the vibration mode of the CO_3_^2−^ ion are caused by the change in the electrostatic valence of the Ca-O bond in the presence of oxygen [89]. At 1114–1081 cm^−1^ and 874–872 cm^−1^ were found absorptions characteristic to the symmetric stretching and out of plane bending modes of CO_3_^2-^ ion. Specific absorption of calcium carbonate in aragonite phase was found at 1114–1081 cm^−1^ and a band that indicates a structural change in the calcium ions from the symmetry of the calcite phase was found at 713–712 cm^−1^ [89].

Some bonds and compounds highlighted by FTIR analysis were the basis of the possible mechanisms proposed for composites degradation presented in Figure 4, Figure 5 and Figure 6. Thus, in Figure 4a,b are proposed two possible degradation mechanisms of natural rubber.

As it can be seen from Figure 4a, by irradiation of the natural rubber matrix, free radicals can be formed by the homolithic cleavage of the double bond. Due to the oxygen presence, the radicals can lead to the formation of aldehydes and ketones by the cleavage of the molecular chain. In addition, carboxylic acids can appear in the presence the formed aldehydes. The polymeric matrix can form radicals in the allylic position by the extraction of an allylic hydrogen from the methyl or methylene group. Unsaturated alcohols can be formed from the allyl radicals also in the presence of oxygen (Figure 4b) [66,74].

Not only the natural rubber matrix but fillers are also affected by the irradiation and can suffer a degradation process. In cellulose, for example, the main component of the organic fillers (sawdust and hemp) the chains are bonded by inter and intramolecular hydrogen bonds. The irradiation of cellulose with doses over 10 kGy, leads to the formation of hydrogen bonds responsible to its degradation.

As the irradiation dose increases, the crystalline structure of the cellulose is altered by cleavage of glycosidic chains [90] and the formation of soluble compounds such as lactones, ketones, levoglucosan or aldonic acid (Figure 5) [91]. The degradation of NR-U and NR-C composites can occur by the cleavage of hydrogen bonds formed between precipitated silica or chalk particles and polymeric matrix, by degradation of the polymer matrix as in Figure 1, but also by the degradation of maleated natural rubber (Figure 6) [92].

FTIR investigation of the four types of composites showed that accelerated electron irradiation of mineral (precipitated silica and chalk) and organic (hemp and sawdust) reinforced composites produced a strong degradation of the latter.

### 3.6. Morphological Investigations through Scanning Electron Microscopy

Morphological changes in the NR-U, NR-C, NR-S and NR-H structures before and after irradiation with 150 and 450 kGy are presented in Figure 6, Figure 7, Figure 8, Figure 9 and Figure 10. In Figure 6 are presented the micrographs of control sample (NR). The homogeneous and smooth surface is characteristic of a material without fillers [89]. The small particles that can be observed even in the non-irradiated sample may come from the vulcanizing additives or may be due to impurities remaining on the surface of the sample after immersion in toluene. The radiation treatment leads to the modification of the surface, especially for the 450 kGy dose, which acquires a rougher and more eroded appearance. During irradiation oxidation and cleavage reactions take place which lead to the formation of low molecular weight compounds, eliminated by immersion in toluene. For this reason, the surface has an eroded appearance and cracks and micro voids are present [93].

In Figure 7 can be observed the non-irradiated sample NR-U that presents a homogeneous and uniform dispersion of precipitated silica [94]. As in the case of the control sample, some agglomerations can be observed which may be due to the impurities remaining on the surface after immersion in toluene. By irradiation, even at 150 kGy but more at 450 kGy, larger agglomerated structures, cracks and micro voids can be observed on the surface, as a result of the degradation process. The formation of small molecule compounds removed by immersion in toluene led to this aspect of the surface after irradiation.

In Figure 8a is presented the micrograph of NR-C sample and a low degree of its dispersion in the polymeric matrix can be observed. The uneven distribution of the chalk has led to the formation of clusters that are visible both for the non-irradiated sample and for the irradiated samples. The surface has a rough appearance and chalk particles (agglomerated or not) seem to peel off the polymer matrix. The agglomeration of CaCO_3_ particles acted as loci of failure that could further decrease the mechanical properties, characteristic for degradation [95].

A comparison of the SEM images of the two mineral fillers (precipitated silica and chalk) shows that precipitated silica has a better particle distribution in the polymer matrix and a better interaction with the polymer matrix (see Table 5).

In Figure 9 and Figure 10 are presented micrographs of non-irradiated and irradiated NR-H and NR-S samples.

The size of the sawdust particles is smaller than that of the hemp particles and this can be seen even in the images of the non-irradiated samples. In addition, the distribution of the sawdust particles is better than that of the hemp. In the micrographs of non-irradiated and irradiate NR-S, the filler particles are not visible, but in the micrographs of NR-H are visible and that can be very well correlated with the results obtained from the evaluation of rubber-filler interaction parameter: 0.794 for NR-S and 0.805 for NR-H at the irradiation dose of 450 kGy. The phenomenon is attributed to the “aging” of the material by oxidative degradation. After irradiation with 150 kGy, no significant differences were observed compared to the non-irradiated sample, in the sense that no voids appeared, and this suggests the continuation of the vulcanization process by irradiation. The increasing of irradiation dose to 450 kGy led to a decrease in crosslinking and gel fraction but also drastic changes in mechanical properties. This means that by increasing the irradiation dose, the degradation process occurs with the formation of short fragments—oligomers that are removed by immersion in toluene. The surface becomes rougher and cracks and micropores appear [96,97]. This result confirms the decrease in adhesion between the filler and the polymer matrix as the irradiation dose increases, leading to degradation of the material by irradiation.

## 4. Conclusions

Degradation of polymeric composites based on natural rubber (NR) reinforced with different fillers by electron beam (EB) irradiation in the dose range of 150 and 450 kGy was investigated. Composites were obtained by peroxide cross-linking in the presence of trimethylolpropane trimethacrylate, the used fillers being natural (sawdust, hemp) and mineral (precipitated silica, chalk) in amount of 50 phr. The degradation process was appreciated by specific analysis: mechanical (hardness, tensile strength, elongation at break), gel fraction, cross-linking degree, uptakes and weight loss variation yields in water and toluene, structural (FTIR) and morphological (SEM). Possible mechanisms specific to degradation are proposed. The cross-linking and the chain scission reaction yield before and after irradiation were appreciated using the Charlesby–Pinner equation. The increasing of *p_0_/q_0_* ratio was associated with the prevalence of scission reaction specific for degradation process as against cross-linking. The change in the studied characteristics of the irradiated samples to values associated with degradation was due to both the increase in the irradiation dose and the type of filler. Degradation was expected to occur as the irradiation dose increased, but the filler type (although used in the same amount of 50 phr) and its specific characteristics led to different results, especially at low irradiation doses were the cross-linking process continued before installing the degradation. Thus, it appears that the cellulose content of the organic fillers and the larger particle size favored an increase in hardness (NR-H at 300 and 450 kGy) and a decrease in tensile strength and elongation at break (NR-S at 450 kGy) while the polar surface chemistry, the smaller particle size of chalk and poor adhesion between the mineral filler and rubber matrix led to values associated with the degradation only in some specific situations (tensile strength of NR-C at 450 kGy). At the same radiation dose of 450 kGy, the sawdust-reinforced composites showed the lowest value of elongation at break. The reinforcing elements in polymer matrix/cellulosic fiber are the unidirectional cellulose microfibers that are responsible with the increasing of stiffness, effect sustained also by the values of cross-linking degree obtained for irradiated and non-irradiated NR-H and NR-S. All composites have presented the same tendency of uptakes and weight loss variation yields in water and toluene associated with degradation, excepting NR-H. FTIR investigation showed that due to the irradiation dose increasing, were found changes in the absorption intensity of specific bands associated with the degradation. A strong degradation has occurred for the composites reinforced with organic fillers. All changes highlighted in the spectra of the irradiated NR-S and NR-H composites, prove the degradation of cellulose, hemicellulose and lignin from hemp and sawdust.

Morphological investigations through SEM showed that by irradiation of the composites with 150 kGy, no significant differences were observed. The increasing of irradiation dose to 450 kGy produced larger agglomerated structures, cracks and micro voids on the surface, as a result of the degradation process. The formation of small molecule compounds removed by immersion in toluene led to this aspect of the surface after irradiation. This is consistent with that the increasing of irradiation dose to 450 kGy led to a decrease in crosslinking and gel fraction but also drastic changes in mechanical properties.

## Data Availability

Not applicable.

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
