# Peer review of "Degradation by Electron Beam Irradiation of Some Composites Based on Natural Rubber Reinforced with Mineral and Organic Fillers"

_ijms, 2022, doi:10.3390/ijms23136925_

Round 1
Reviewer 1 Report
Dear Editor
This manuscript “Degradation by electron beam irradiation of some composites based on natural rubber reinforced with mineral and organic fillers” by Elena et. al. demonstrated the effect of electron beam on the composites of natural rubber and mineral/organic filers produced the larger agglomerated structures, cracks and micro voids on the surface, resulted into the degradation of the composites.
The authors looks to completely reflect the previous reviewer’s comments and the manuscript should be improved. Overall, this manuscript has well organized structures and logic flow to persuade their results using proper figures. However, it is still needed the correction of English to clearly transmit their idea. Based on the careful review, I recommend this manuscript to be published in the IJMS.
Author Response
English style has been improved

Reviewer 2 Report
ijms-1766702
Degradation by electron beam irradiation of some composites based on natural rubber reinforced with mineral and organic fillers
Elena Manaila, Gabriela Craciun, Daniel Ighigeanu, Ion Bogdan Lungu, Marius Dumitru, Stelescu Maria Daniela
The work is about the reinforcement of natural rubber with fillers and the resistance of the rubber in irradiation. It was found that irradiation dose to 450 kGy produced larger agglomerated structures, cracks and micro voids on the surface. It also leads to a decrease in crosslinking and gel fraction but also drastic changes in mechanical properties specific to the composites degradation process. In the case of the composites reinforced with mineral fillers the degradation can occur by the cleavage of hydrogen bonds formed between precipitated silica or chalk particles and polymeric matrix also.
The article seems to be the revised version of a manuscript. I believe that the manuscript can be accepted after minor revision.
Please consider to move equations (7) and (8) with the corresponding text to Experimental Section
Change the (,) with (.) in the numbers in axes. (see for example Figure 3)
Please check the English. Small changes can be performed.
Please follow the journal format for references.
Some references can be deleted like 5, 80 etc.
In values presented in tables keep the correct number of decimal points. For example 3.2999 ± 0.14 if the deviation is ± 0.14 then the value should be 3.30. If the value is 3.2999 then the deviation should be 0.000XX or even better 0.0000X
Author Response
Accordingly to the reviewers and editor requests the changes that have been made in the manuscript are marked in track changes mode.
Thus,
- English style has been improved in accordance with the suggestions that have been done by both Reviewers;
- In the Introduction a paragraph has been rewritten in accordance with the suggestions that have been done by Reviewer 2;
- In the subchapter 2.2. Samples irradiation, some modifications of irradiation parameters have been done;
- A new subchapter named 2.3.4. Rubber- filler interaction, containing the equations (7) and (8), was introduced and as a consequence all the equations have been renumbered in accordance with the suggestions that have been done by Reviewer 2. From this point forward, the references have been renumbered.
- The data presented in Table 4 were modified in accordance with the suggestions that have been done by Reviewer 2;
- Figure 3 has been modified in accordance with the suggestions that have been done by Reviewer 2;
- All the references are relevant to the contents of the manuscript and as a consequence none has been removed.
